# Exploiting Time Series of Sentinel-1 and Sentinel-2 Imagery to Detect Meadow Phenology in Mountain Regions

**Laura Stendardi** [1,*] , **Stein Rune Karlsen** [2], **Georg Niedrist** [3], **Renato Gerdol** [4] , **Marc Zebisch** [5], **Mattia Rossi** [5] **and Claudia Notarnicola** [5]

1    Faculty of Science and Technology, Free University of Bozen-Bolzano, 39100 Bolzano, Italy
2    NORCE Norwegian Research Centre AS, Measurement Science, P.O. Box 6434,
     N-9294 Tromsø, Norway; skar@norceresearch.no
3    EuracResearch, Institute for Alpine Environment, 39100 Bolzano, Italy; georg.niedrist@eurac.edu
4    Department of Life Sciences and Biotechnology, University of Ferrara, I-44121 Ferrara, Italy; grn@unife.it
5    EuracResearch, Institute for Earth Observation, 39100 Bolzano, Italy; marc.zebisch@eurac.edu (M.Z.);
     mattia.rossi@eurac.edu (M.R.); claudia.notarnicola@eurac.edu (C.N.)
*    Correspondence: Laura.Stendardi@natec.unibz.it

**Abstract:** A synergic integration of Synthetic Aperture Radar (SAR) and optical time series offers an unprecedented opportunity in vegetation phenology monitoring for mountain agriculture management. In this paper, we performed a correlation analysis of radar signal to vegetation and soil conditions by using a time series of Sentinel-1 C-band dual-polarized (VV and VH) SAR images acquired in the South Tyrol region (Italy) from October 2014 to September 2016. Together with Sentinel-1 images, we exploited corresponding Sentinel-2 images and ground measurements. Results show that Sentinel-1 cross-polarized VH backscattering coefficients have a strong vegetation contribution and are well correlated with the Normalized Difference Vegetation Index (NDVI) values retrieved from optical sensors, thus allowing the extraction of meadow phenological phases. Particularly for the Start Of Season (SOS) at low altitudes, the mean difference in days between Sentinel-1 and ground sensors is compatible with the acquisition time of the SAR sensor. However, the results show a decrease in accuracy with increasing altitude. The same trend is observed for senescence. The main outcomes of our investigations in terms of inter-satellite comparison show that Sentinel-1 is less effective than Sentinel-2 in detecting the SOS. At the same time, Sentinel-1 is as robust as Sentinel-2 in defining mowing events. Our study shows that SAR-Optical data integration is a promising approach for phenology detection in mountain regions.

**Keywords:** Sentinel-1 and Sentinel-2; time series analysis; start of season, harvest, mountain region

## 1. Introduction

Agricultural management in European mountain regions is a key strategy for preserving ecosystem stability and regional economies [1,2]. Phenology is defined as "the study of the timing of recurring biological events, the causes of their timing regarding biotic and abiotic forces, and the interrelation among phases of the same or different species" [3].Vegetation phenology is a relevant indicator of crop productivity and health. Phenological stage monitoring is therefore crucial in the decision-making process of the agricultural management [4]. In mountainous regions, agricultural areas are generally of small size and the vegetation is characterized by a heterogeneous distribution. In addition, mountain crops are vulnerable to climate variability [5–7]. Satellite imagery plays a unique and important role in monitoring crop and soil conditions for farm management [8–10]. In the past

years, most studies using satellite imagery for crop and natural vegetation monitoring have focused on the use of optical imagery. By exploiting the reflectance of visible and Near Infra-Red (NIR) radiation and the emittance of thermal Infra-Red (IR) radiation, canopy characteristics have been mapped over large areas [11]. The Normalized Difference Vegetation Index (NDVI) [12] has been widely used to detect phenological phases [13–18]. In this case, cloud contamination and topographic effects in mountain regions compromise data significantly in the optical domain [19].

Microwave wavelengths have important advantages over optical remote sensing for agricultural applications, because they pass through the atmosphere and clouds with negligible attenuation [20]. This allows frequent measurements over the short growing season of mountain crops. Conversely, the radar signal can be difficult to interpret as the total radar backscatter is a complex sum of the backscatter from vegetation and soil. The radar beam can penetrate both the canopy and soil to a difficult-to-determine depth, making it complicated to determine if the signal is dominated by either vegetation or soil conditions [21,22].

Reliable ground measurements of crop growth stages and soil moisture throughout the growing season are therefore important to understand the relative influence of these factors on the microwave signal. Additionally, dense time series are necessary to understand the Synthetic Aperture Radar (SAR) signal behavior with regards to crops. From the first attempt to monitor rice crops [23], relevant results were found by combining different SAR sensors [24], incidence angles [25], different polarizations [26–31], and Interferometric SAR technique [32]. A data fusion approach was developed using a dynamical framework based on particle filter (PF). This approach has shown that the incorporation of additional sources to the NDVI time series can improve the phenological monitoring. The inclusion of SAR images in particular increases the sensitivity to crop dynamical development and improves results in the process of estimating specific phenological states [33]. Furthermore, the grassland mowing event detection were explored by applying coherence estimation on interferometric acquisitions [34] and using radar polarimetry [35,36]. Crop structure, dielectric properties of the canopy, soil roughness, and moisture influence the backscattering coefficients. Moreover, the crop structure and plant water content vary depending on phenological stages and crop condition. With multipolarization, it is possible to explore the sensitivity of waves to different orientation, shape and dielectric properties of elements in the scattering field [37]. Both the HH and VV polarizations operating in C-band are sensitive to soil moisture variations, whereas the cross-polarized backscatter is primarily associated with volume scattering of vegetation [38]. The different attenuation of VV and HH polarization is useful for discriminating crop types and the cross-polarized channel with a higher dynamic can improve the crop separability. Moreover, grassland and crop discrimination is achievable by using multitemporal SAR images [39]. Also, for phenology and its parameters, the cross-polarized channel gives a higher contrast between high and low productivities [38,40]. The trends in radar backscatter, measured on different dates, can be correlated with soil moisture content, since the effects of spatial roughness variations are smoothed [41]. To reduce these factors, [42] suggested that the ratio of backscatter measured on two close successive dates might be a simple and effective way to decouple the effect of vegetation and surface roughness from the effect of soil moisture changes, when volumetric scattering by the crop canopy is not dominant.

For robust retrieval methods, the temporal change of backscattering coefficients on mountain ecosystems still needs to be documented. Moreover, an integration of multisensor time series has to be evaluated on meadow phenology detection.

Within the Copernicus programme we now have the possibility to explore different sensors. Both Sentinel-1A and 1B satellites with their SAR sensors provide time series of medium and high resolution of C-band data [43], simultaneously Sentinel-2A and 2B optical sensors acquire 13 spectral bands in the visible, the NIR, and the Short Wave IR (SWIR) [44]. Combining the two Sentinel satellites with a revisiting time of six and five days, respectively, offers an unprecedented opportunity to monitor crop in mountain regions.

In this study, we analyzed time series from the Sentinel-1 (S-1) and Sentinel-2 (S-2) together with proximal sensors to understand their temporal behavior for mountain meadow areas. The main objectives of this paper are: (1) to understand and quantify the impact on multi temporal SAR images of different grassland types and soil conditions in the perspective of data integration; (2) exploit the synergic use of SAR and optical data to retrieve maps of mountain phenology (start of the season and harvesting time).

With respect to the above presented studies, the novelties of this work are:

1. Detection of phenological stages of meadows in mountain ecosystem using multitemporal SAR imagery;
2. Mapping of phenology with SAR data using a statistical approach.

For the first time, S-1 and S-2 are evaluated in synergy in the phenological retrieval process. A multisensor methodology is presented and compared to establish a common and complementary approach for the detection of mountain phenology.

## 2. Study Area and Datasets

### 2.1. Study Area

The study area is the South Tyrol region located in northern Italy (Figure 1). South Tyrol has an area of 7400 km$^2$ and is situated in the center of the Alps with steep elevation gradients stretching from 190–3890 m a.s.l. Typical agricultural land-use types are meadows, pastures, orchards, and vineyards. Around 79% of the region is above 1200 m a.s.l., with small valley floor surfaces and steep slopes. Moreover, around 50% (3228 km$^2$) of South Tyrol is covered by forest and about 30% is used for agriculture [45,46].

Within the MONALISA project (http://www.monalisa-project.eu/en/home/Pages/default.aspx) several environmental stations were installed in the area with the main aim to monitor vegetation and soil properties. Figure 1 shows the land-cover types, the elevation of the area and the stations used in our study as ground reference. Each name of the station includes information about location, vegetation cover type, and slope of the area. Moreover, the names include the altitude of the stations. For a comprehensive description of the acronyms see Table 1. The list of the stations can be viewed at the following website: http://monalisasos.eurac.edu/sos/static/client/helgoland/index.html#/map.

**Table 1.** Acronyms of ground stations used in this study.

| Ground Station | Acronym |
|---|---|
| **domef 1500** | do = Dolomites, me = meadow, f= flat, 1500 = 1500 m a.s.l. |
| **domef 2000** | do = Dolomites, me = meadow, f= flat, 2000 = 2000 m a.s.l. |
| **vimef 2000** | vi = Vinschgau/Venosta valley, me = meadow, f= flat, 2000 = 2000 m a.s.l. |
| **vimes 1500** | vi = Vinschgau/Venosta valley, me = meadow, s= steep, 2000 = 2000 m a.s.l. |

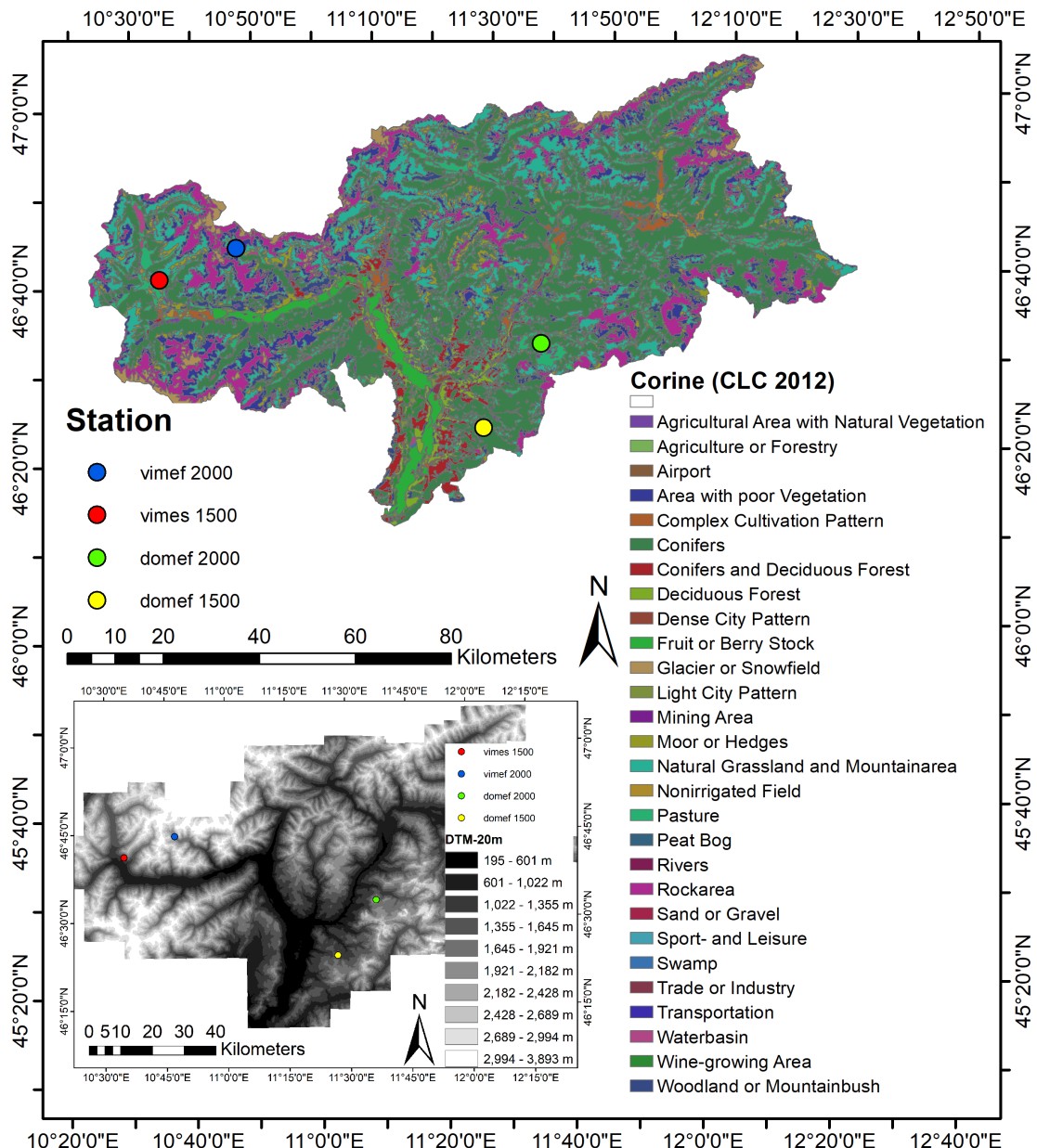

**Figure 1.** Corine Land-cover map (CLC 2012) of South Tyrol and Digital Terrain Model (DTM 20 m). On each map we overlaid four ground stations.

## 2.2. Datasets

The analyzed data sets are composed of 59 S-1A images acquired from October 2014 to September 2016. The data belongs to track 168, which covers the region almost entirely, every 12 days. The 78 S-2A images span from June 2015 to November 2016, with a temporal coverage of 10 days. Although the number of S-2 images is higher than S-1, the presence of clouds reduced the data availability in our time series. Clouds generated missing data in the optical images especially during the winter period and at high altitudes. Out of 78 S-2 scenes we were able to use an average of 39 scenes in the areas at 1500 m and 31 images in the areas at 2000 m a.s.l. On the contrary, in the microwave domain the images were consistent during the seasons, except for the missed acquisitions caused by onboard satellite problems. S-1A instrument was unavailable during 2016, between 8 June and 14 July [47]. Together with satellites images, ground data were available for meadow and pasture areas. Most

of the stations are equipped with sensors providing information on Soil Water Content (SWC), Soil Temperature (ST), and NDVI-PRI from the Spectral Reflectance Sensor (SRS, Decagon Devices Inc., Pullman, WA, USA). The data have been recorded since 2015 with a time step of 15 min. SWC is available at different depths from 2 to 20 cm (Soil Water Content Reflectometer, Campbell Scientific, Edmonton, AB, Canada). A few of these are further equipped with PhenoCams acquiring both an RGB and a combined RGB + IR image in 30-min intervals with $1296 \times 960$-pixel resolution. The cameras used were StarDot Hybrid IP 1.3 Megapixel Netcams (StarDot Technology, Buena Park, CA, USA) mounted atop the MONALISA stations in 2015.

In this study, we selected four representative stations, fully equipped with SWC, ST at 2, 5, 20 cm, NDVI & PRI and PhenoCam and located at 1500 and 2000 m a.s.l. For the detailed description of the four selected ground stations see Table 2.

**Table 2.** List of the stations and related measurements available: SWC (Soil Water Content), ST (SoiL Temperature), PAR (Photosynthetic Active Radiation), NDVI (Normalized Difference Vegetation Index), PRI (Photochemical Reflectance Index).

| Ground Station | Parameters | Latitude | Longitude | Altitude |
|---|---|---|---|---|
| **domef 1500** | SWC & ST 2, 5, 20 cm, PAR, NDVI & PRI, PhenoCam | 46.401002 | 11.454211 | 1500 m a.s.l. |
| **domef 2000** | SWC & ST 2, 5, 20 cm, PAR, NDVI & PRI, PhenoCam | 46.556687 | 11.614836 | 2000 m a.s.l |
| **vimef 2000** | SWC & ST 2, 5, 20 cm, PAR, NDVI & PRI, PhenoCam | 46.745151 | 10.788845 | 2000 m a.s.l |
| **vimes 1500** | SWC & ST 2, 5, 20 cm, NDVI & PRI, PhenoCam | 46.686163 | 10.579881 | 1500 m a.s.l |

## 3. Methodology

The overall scheme of the proposed methodology is illustrated in Figure 2. The central aim of the procedure is to derive from time series of S-1 and S-2 images the main phenological features. The whole procedure is divided into four main steps. After a preprocessing of the S-1 and S-2 images, they are co-registered as to refer to the same area of interest. Then the S-1 and S-2 time series are extracted over the selected areas where also ground data are available. Subsequently, the backscatter from S-1 and the NDVI from S-2 and ground sensors are modeled to extract the main features of the phenocycle such as start of the season and mowing event. Finally, maps are produced, and the validation is carried out. In the following, each step of the procedure is described in detail.

### 3.1. Preprocessing of S-1, S-2 Images and Ground Observation

The S-1 data preprocessing encompasses several standard steps to derive geocoded intensity images starting from the Ground Range Detected (GRD) data.

These operations were performed using the tools provided by SNAP (Sentinel Application Platform) and custom algorithms developed in Python. Beside the standard operations, a spatial and temporal speckle filter was used [38]. The S-2 images were preprocessed using the Sen2Cor processor (v.2.3) without cirrus or topographic correction. All non-vegetated areas were masked using the CORINE 2012 (CLC 2012) land-cover information. Both S-1 and S-2 data were corrected to eliminate layover/shadow zones and to reduce the contamination due to cloud presence based on the Sen2Cor scene classification, respectively. The values recorded by the ground sensors were averaged according to intervals of time, averaging four values each interval. The SRS sensor has wavebands centered at 650 nm (Red) and 810 nm (NIR) [48]. To calculate the NDVI, we used the formula [12]:

$$\text{NDVI} = \frac{\text{NIR} - \text{Red}}{\text{NIR} + \text{Red}} \tag{1}$$

Subsequently, we calculated average of measurements around 10:00 a.m. (Sentinel-2 acquisition time, UTC time zone). To define the start, maximum and the end of the growing season at the four stations we did a visual analysis of the images from the PhenoCams. We used a common protocol

as in [49], defining the Start Of the growing Season (SOS) as 50% green leaves developed, time of maximum (MAX) as full-size leaves and the End Of the growing Season (EOS) as 50% yellow leaves. In addition, we recorded the harvesting time.

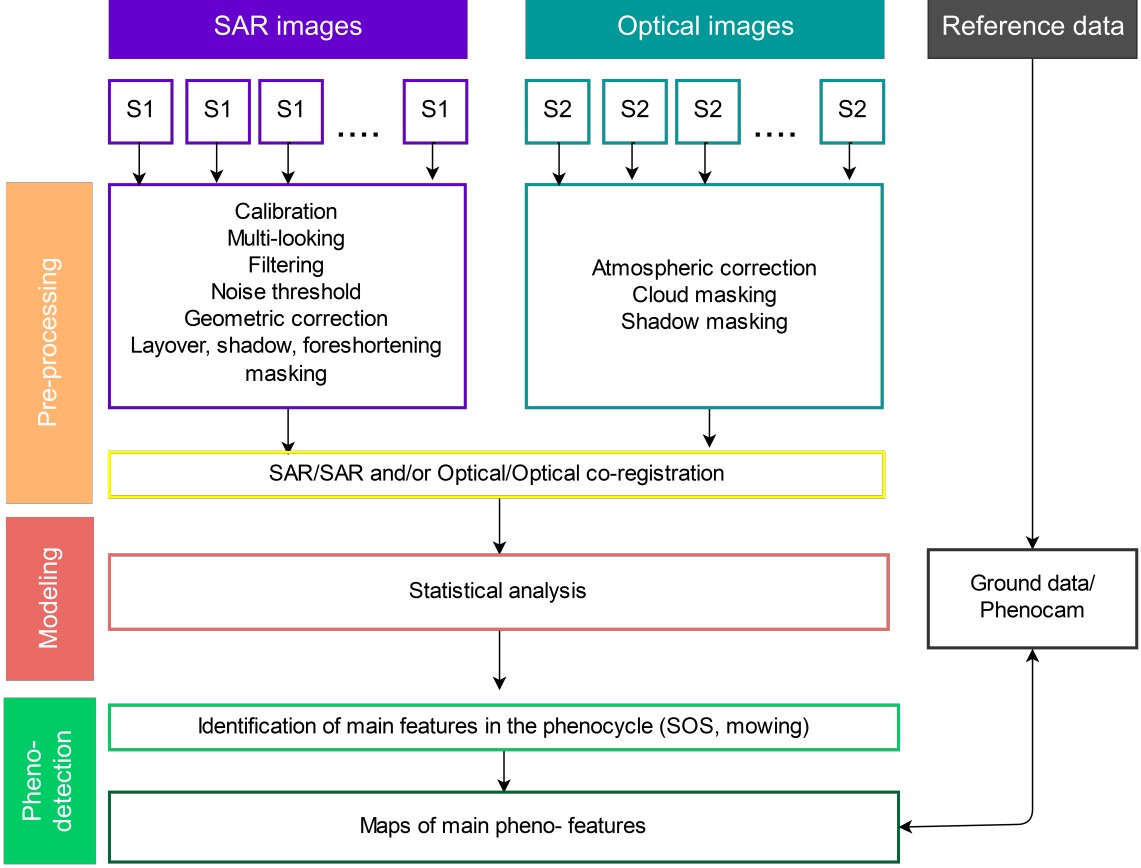

**Figure 2.** Flow chart of the proposed approach to detect the main features of the phenocycle for the selected areas.

## 3.2. Statistical and Electromagnetic Modeling

The correlation and modeling analyses are a preparatory phase to detect the phenocycle features from S-1 time series. The aim is to understand the temporal behavior of S-1 imagery and correlate this trend to soil and vegetation parameters. Even though the current study focuses on meadows phenology, as a first step, S-1 backscattering coefficients in VV and VH polarization were extracted over different crops and land-cover types to understand the radar signal dynamics to different vegetation types. A stack of images has been created and, considering a land-cover map of the area (CORINE Land Cover 2012), several regions of interest (around 10 for each land-cover type) were extracted for the following land-cover classes: meadow, pasture, orchard, vineyard, and cereal. Coniferous and deciduous forest have also been considered for the sake of comparison with the other classes. The size of each area of interest (around 200 m$^2$) was selected as a compromise between homogeneity of the area and number of pixels. Trends in VH and VV were analyzed for the different land-cover classes.

Subsequently, time series of S-1, in areas of interest corresponding to the ground stations, were compared with other sources of information. In detail, they are compared with time series of NDVI derived from S-2 and SRS, and with soil parameters such as ST and SWC. This analysis is carried out for meadows, which are the target land-cover type observed by the ground stations. The analyses were done for the season 2016, considering the data availability of the different sensors. For comparing the backscatter with point data, the areas of interest were extracted close to the station, in homogeneous areas. After obtaining time series of values from the selected areas, a -correlation between both

optical and SAR sensors was performed and noise-reduction filters were applied depending on the investigation purpose. The cross-correlation analysis was performed by using the t-series v0.1-2 package [50] on R (v. 3.4.3.). After with analyzed the SAR signal for different land-cover types, as a last step, a simulation with electromagnetic models were carried out, specifically for meadows. The simulation was performed to understand in a quantitative way the impact of the different parameters (soil and vegetation) on the SAR signal.

The total scattering from vegetated soils was simulated by using the Water Cloud Model (WCM) [22]:

$$\sigma^0_{pq,tot} = \frac{A\cos(\theta)}{2b\mathrm{NDVI}}(1 - \exp(-2b\mathrm{NDVI}\sec(\theta)) + \sigma^0_{pq,b}\exp(-2b\mathrm{NDVI}\sec(\theta)) \tag{2}$$

where in Equation (1) the dependence on the vegetation is expressed through NDVI as a proxy of vegetation water content; $\theta$ is the local incidence angle, $\sigma^0_{pq}$ is the scattering from bare soil that for the VH polarization was simulated with the Oh model [51]. A and b are parameters for crop type and were fitted against ground data. A range of crop parameters were tested to determine the best-fitting combination. The simulated backscatter was analyzed through linear regressions. Subsequently, the influence of the signal components were examined through a dominant factor analysis [52].

### 3.3. Phenological Phases Extraction

A Best Index Slope Extraction (BISE) [53] filter was applied on optical and SAR time series to extract phenological parameters; BISE filter was chosen to remove the noisy points affected by the mowing that could interfere with the detection of annual vegetation cycle. Then, four filter techniques were tested: Savitzky Golay, Double-logistic, Linear Filter and Fast Fourier Transform (FFT). For each of the modeled time series we extracted the SOS and EOS, with a threshold of 50% and the maximum (MAX) of the curve [15]. The analyses were performed using the Phenex package [54] on R (v. 3.4.3.). Then, the results were validated by using the phenological information derived by PhenoCams and SRS sensors. Subsequently, a phenological map of the SOS was created as a final product for the season 2016, using a Linear Filter both for SAR and optical images, on codes developed in Python (v. 2.7). To detect the SOS we applied a pixel-by-pixel-based method used in [55] and discussed in [56]. The optical map was created as a reference for the SAR one.

Conversely, to identify the harvest time, a linear filter was applied. This filter allowed to preserve unaltered seasonal trend of a given time series. Based on the maximum (MAX), obtained from the previous analysis, a minimum between intervals of time was used to detect the first and, eventually, the second mowing. Next, S-1 and S-2 maps of the harvest time were produced and compared to PhenoCam images through a visual interpretation, to detect the date of mowing.

## 4. Results

First, the trends of VV and VH polarization are obtained over different crop types. Next, the capability of the WCM to reproduce vegetation characteristics is evaluated and discussed. Then, in the selected areas, the phenological phases extraction is performed both for S-1 and S-2 time series and compared with the observations of the fixed stations. Finally, phenological maps of SOS and harvesting time are produced on large scale.

### 4.1. Statistical and Electromagnetic Modeling

Both VH and VV polarization signals show a strong correlation in C-band for all different land-use types, as a result of the high sensitivity to vegetation biomass with respect to SWC. The highest $\sigma^0$ values in both polarizations were associated with the period in which the crop green biomass generally reaches its maximum. The trends belonging to vineyards, orchards, and deciduous forest show a higher level of the signal, ranging from $-11.5$ dB to $-7.5$ dB and from $-18.5$ dB to $-13.5$ dB for VV and VH polarization, respectively. Forests of conifer show a lower signature in both polarizations. Cereals

present a similar trend in the VV and VH channel, while meadows and pastures exhibit seasonal changes in the dynamic range, depending on the polarization. In the VV channel, pastures show a high dynamic range, from −14 dB to −6.5 dB, with a peak in November and lower values during the summer due to the scarcity of water, which was particularly strong in 2015 (http://weather.provinz.bz. it/historical-data.asp). Conversely, in the VH polarization the $\sigma^0$ values demonstrate less sensitivity to seasonal dynamics, with value ranging from −17 dB to −21 dB. The signature of meadows, which are strongly managed in terms of fertilization, irrigation, and mowing in this area, varies between −14.5 dB and −8.5 dB and from −24.5 dB and −15.5 dB for VV and VH polarization, respectively. In the VH polarization, the trend of meadows shows the highest dynamic and is clearly distinguishable from pastures and from the other classes. Figure 3 illustrates an example of the S-1 backscatter trend for different land-use types. The smoothing lines show a local polynomial regression fitting (loess), done using neighboring values, weighted by their distance to the point [57].

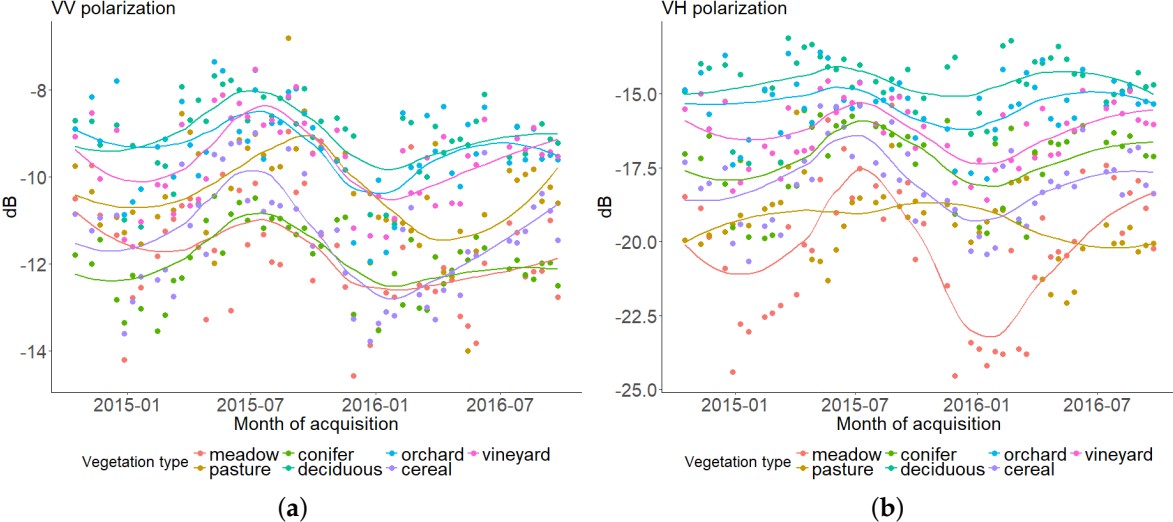

**Figure 3.** Trends of S-1 backscatter for different land-use types: orchards, meadows, crops, deciduous forest, coniferous forest, and pastures. (**a**) VV polarization over the area of interest. (**b**) VH polarization over the area of interest. The smoothing lines are obtained with a local polynomial regression fitting.

The relationship between Sentinel-2 NDVI and backscattering coefficients was compared for four meadow areas with NDVI measured at the ground (SRS sensor). The strongest correlation between S-1 and S-2 in semi-natural habitats was found in the VH channel ($R^2 = 0.52$).

For illustration purposes, the results are presented for a single area, while Tables 3 and 4 summarizes the statistics. Figure 4a,b illustrates time series of sensors in the area of vimef 2000, where a similar trend is visible during the entire acquisition period with an increase of VH signal of around 4–6 dB in the summer period as the NDVI advances from 0.5 to 0.8 after the snow melting. The NDVI from Sentinel-2 shows a higher value during all the temporal profile compared to the ground sensors, with values around 0.8–0.9, during the summer peak. Figure 4c illustrates a similar trend between $\sigma_{VH}$ and temperature, while Figure 4d demonstrates a shift in the lag between SWC and scattering coefficients.

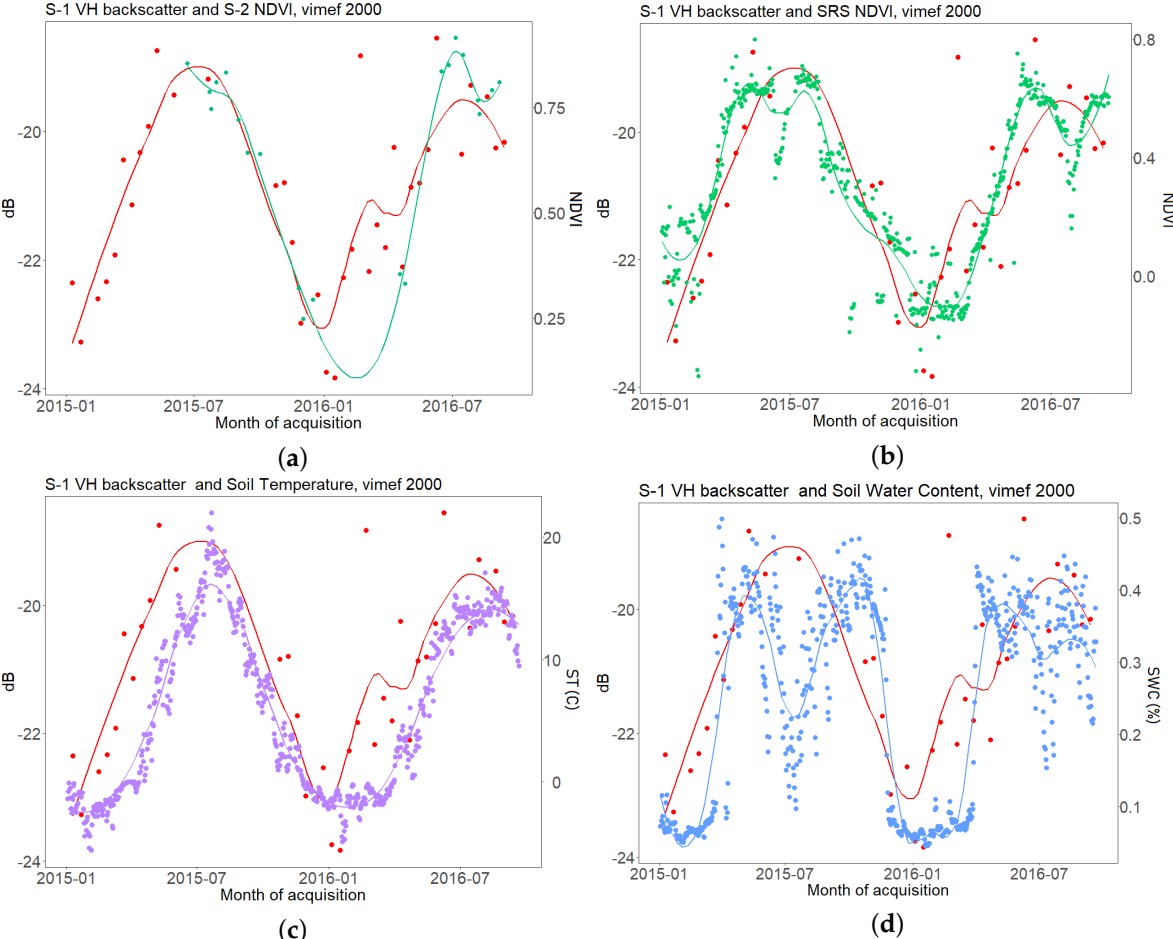

**Figure 4.** Temporal evolution of the mean backscattering coefficient VH (red)—Sentinel-2 NDVI (dark green) (**a**), VH (red)—NDVI from the SRS ground sensor (green) (**b**), VH (red)—soil temperature at 2 cm (violet) (**c**) and VH (red)—SWC (blue) (**d**), for the area domef 1500, during the acquisition period 2015–2016.

To understand the evolution of the VH signal in respect of the NDVI of both S-2 and SRS sensors, a temporal analysis of the season 2016 was computed. First, we analyzed the interaction between these three series of data through a cross-correlation function. The correlation was positive between sensors in all the areas. As shown in Table 3, for the areas of domef 2000 and vimes 1500 there is a minor shift in lags (h) for all the cross correlations. Conversely, the most dominant cross correlations in the area domef 1500 occur between lags −10 and −12 among S-1 and S-2, and lags 19 and 22 among S-1 and SRS. Furthermore, due to a high cloud cover contamination in the S-2 time series, the vimef 2000 area shows a shift in the lag both for the S-2/SRS and the S-1/S-2 correlation. The most dominant cross correlations occurred instead between lags −1 and 1 for S-1/SRS.

Additionally, Table 3 demonstrates a high Pearson's product-moment correlation between each sensor. There is a positive relationship, in the range 0.69 to 0.84, between Sentinel-1 $\sigma_{VH}$ and Sentinel-2 NDVI, and a strong positive relationship between both the NDVI (from 0.54 to 0.69) as well as between SRS and Sentinel-1 (from 0.45 to 0.88). Due to the cloud contamination at high altitude in the optical domain, the Pearson's product-moment produces higher values between SAR and ground sensor time series in the areas domef 2000 and vimef 2000.

**Table 3.** Pearson's product-moment correlation and the most dominant cross correlations at lags = h, between S-1 VH backscatter, S-2 NDVI and SRS NDVI of the areas of interest.

| Ground Station | S-2/SRS (Person Correlation and Lags) | S-1/S-2 (Person Correlation and Lags) | S1/SRS (Person Correlation and Lags) |
|---|---|---|---|
| domef 1500 | 0.54, h = [−2, 0] | 0.70, h = [−10, −12] | 0.51, h = [19, 22] |
| domef 2000 | 0.69, h = [−5, −3] | 0.80, h = [−6, −4] | 0.88, h = [−3, 0] |
| vimes 1500 | 0.66, h = [−1, 1] | 0.84, h = [−2, 0] | 0.45, h =[−2, 0] |
| vimef 2000 | 0.57, h = [13, 15] | 0.69, h = [6, 8] | 0.71, h = [−1, 1] |

Based on the output of our analysis, $\sigma^0$ VH time series follow the trend of the NDVI during all seasons in the selected areas thus indicating the possibility of phenological phases extraction. To complete the analysis, we investigated the impact of soil and vegetation on this signal for meadows, through simulations with the WCM. The crop parameters have been fitted to achieve the best match between backscatter coefficients and WCM. After testing a permutation of A and b variables (100 samples each) in Equation (1), $A = 0.001$ and $b = 0.002$ were the combination that best fitted the WCM for meadows. The simulated backscatter from vegetated soil follows the trend of the measured $\sigma_{VH}$ in domef 1500 (Figure 5a) and vimes 2000 (Figure 5b). Conversely, the poor result of vimes 1500 (Figure 5c) is due to problems in the ground data acquisition. For the season 2015 the NDVI reaches a saturation on July 20 and remains with values around 0.98 until December. Therefore, this station cannot be involved in the analysis of the results. Moreover, the station domef 2000 starts to acquire the ground NDVI only from May 2015. This reduces the output of the simulation (Figure 5d).

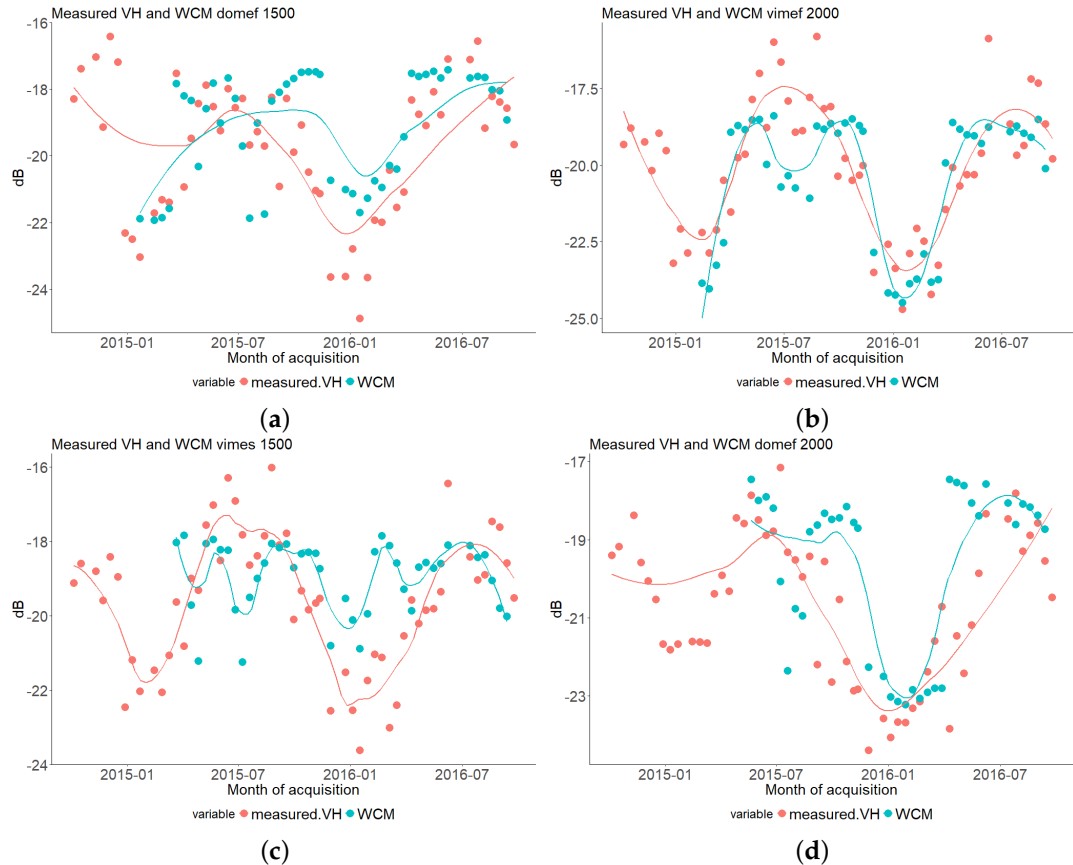

**Figure 5.** Measured VH backscatter and simulation through the WCM: the four panels show the stations domef 1500 (**a**), vimef 2000 (**b**), vimes 1500 (**c**) and domef 2000 (**d**). The smoothing lines are obtained with a local polynomial regression fitting.

Table 4 summarizes the results of model fitting between $\sigma_{VH}$ and different parameters (WCM, NDVI and SWC) for the four areas. The linear regression between simulation and VH signal has an adjusted $R^2$ of 0.52 and 0.55 for the stations domef 1500 and vimes 2000, respectively. In addition, the Root-Mean-Square Error (RMSE) corresponds to 1.40 dB and 1.64 dB. The relation between the model and the $\sigma_{VH}$ drops to an $R^2$ of 0.23 in domef 2000 station, while the RMSE rises to 2.40 dB. The ground NDVI shows on average a higher adjusted $R^2$ ($R^2$ mean of 0.6) in the linear model with the VH backscatter, than the SWC ($R^2$ mean of 0.37). By comparing the resulted linear relations, through the method of dominant factors, the influence of vegetation expressed in terms of NDVI on the WCM is greater than the soil component.

**Table 4.** Summaries of the results of linear model fitting between parameters.

| Linear Model | Ground Station | Adjusted $R^2$ | $p$-Value (<) | RMSE (dB) |
|---|---|---|---|---|
| VH~WCM | domef 1500 | 0.52 | $1.369 \times 10^{-8}$ | 1.40 |
| VH~WCM | domef 2000 | 0.23 | 0.001 | 2.40 |
| VH~WCM | vimes 1500 | 0.04 | 0.096 | 1.90 |
| VH~WCM | vimef 2000 | 0.55 | $3.39 \times 10^{-9}$ | 1.64 |
| VH~NDVI | domef 1500 | 0.32 | $2.20 \times 10^{-5}$ | |
| VH~NDVI | domef 2000 | 0.68 | $1.15 \times 10^{-10}$ | |
| VH~NDVI | vimes 1500 | 0.05 | 0.071 | |
| VH~NDVI | vimef 2000 | 0.80 | $2.2 \times 10^{-16}$ | |
| VH~SWC | domef 1500 | 0.47 | $2.82 \times 10^{-8}$ | |
| VH~SWC | domef 2000 | 0.17 | 0.006 | |
| VH~SWC | vimes 1500 | 0.04 | 0.096 | |
| VH~SWC | vimef 2000 | 0.47 | $9.24 \times 10^{-9}$ | |

## 4.2. Phenological Phases Extraction

### 4.2.1. SOS

The retrieved phenological phases from the four areas are compared with the information extracted from the PhenoCams in the field. Figure 6 shows an example of the SAR modeled time series with the phases of SOS (red line), MAX (yellow line) and EOS (blue line). Meanwhile Table 5 summarizes the results, expressed in Day Of Year (DOY), of sensors and PhenoCams.

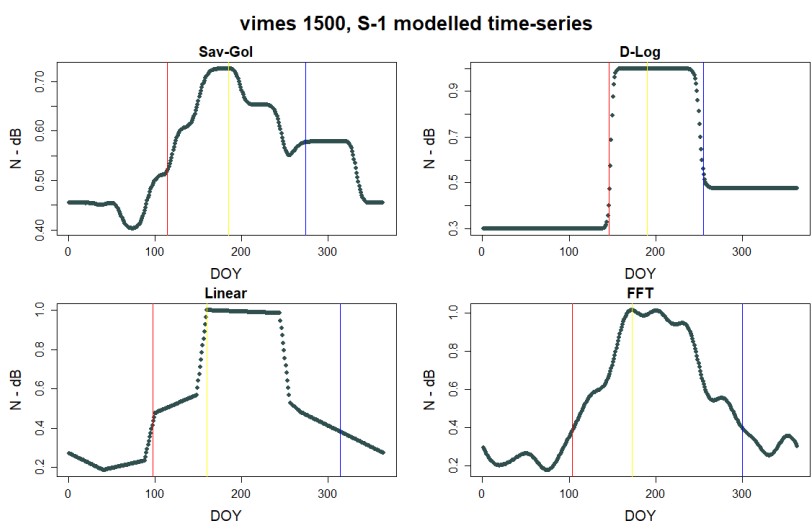

**Figure 6.** Example of modeled time series using Savitzky Golay, Double-logistic, Linear, and FFT filters. The figure shows the phenocycle phases extraction in vimes 1500 area using the normalized S-1 VH backscatter (N-dB).

**Table 5.** Day Of Year (DOY) of phenological phases for all the filtered time series. In bold the DOY extracted from PhenoCams.

| Area | Phenophase | Filter | Phenocam (DOY) | S-1 (DOY) | S-2 (DOY) | SRS (DOY) |
|------|-----------|--------|----------------|-----------|-----------|-----------|
| domef 1500 | SOS | | **102** | | | |
| | SOS | Sav-Gol | | 66 | 106 | 106 |
| | SOS | D-Log | | 151 | 101 | 101 |
| | SOS | Linear | | 90 | 76 | 105 |
| | SOS | FFT | | 94 | 82 | 106 |
| | MAX | | **185** | | | |
| | MAX | Sav-Gol | | 191 | 246 | 241 |
| | MAX | D-Log | | 186 | 198 | 111 |
| | MAX | Linear | | 208 | 245 | 115 |
| | MAX | FFT | | 290 | 247 | 127 |
| | EOS | | **301** | | | |
| | EOS | Sav-Gol | | 246 | 307 | 301 |
| | EOS | D-Log | | 265 | 317 | 312 |
| | EOS | Linear | | 259 | 306 | 218 |
| | EOS | FFT | | 253 | 309 | 236 |
| domef 2000 | SOS | | **147** | | | |
| | SOS | Sav-Gol | | 96 | 38 | 149 |
| | SOS | D-Log | | 64 | 90 | 151 |
| | SOS | Linear | | 67 | 92 | 23 |
| | SOS | FFT | | 75 | 82 | 79 |
| | MAX | | **205** | | | |
| | MAX | Sav-Gol | | 230 | 124 | 242 |
| | MAX | D-Log | | 229 | 124 | 201 |
| | MAX | Linear | | 208 | 189 | 166 |
| | MAX | FFT | | 196 | 145 | 272 |
| | EOS | | **282** | | | |
| | EOS | Sav-Gol | | 298 | 230 | 284 |
| | EOS | D-Log | | 260 | 230 | 288 |
| | EOS | Linear | | 318 | 270 | 284 |
| | EOS | FFT | | 318 | 252 | 289 |
| vimes 1500 | SOS | | **99** | | | |
| | SOS | Sav-Gol | | 127 | 95 | 96 |
| | SOS | D-Log | | 147 | 96 | 96 |
| | SOS | Linear | | 89 | 97 | 97 |
| | SOS | FFT | | 81 | 104 | 97 |
| | MAX | | **180** | | | |
| | MAX | Sav-Gol | | 179 | 210 | 137 |
| | MAX | D-Log | | 164 | 176 | 261 |
| | MAX | Linear | | 160 | 219 | 175 |
| | MAX | FFT | | 181 | 229 | 139 |
| | EOS | | **306** | | | |
| | EOS | Sav-Gol | | 265 | 309 | 313 |
| | EOS | D-Log | | 306 | 300 | 330 |
| | EOS | Linear | | 309 | 295 | 327 |
| | EOS | FFT | | 303 | 282 | 290 |
| vimef 2000 | SOS | | **140** | | | |
| | SOS | Sav-Gol | | 122 | 154 | 128 |
| | SOS | D-Log | | 154 | 162 | 131 |
| | SOS | Linear | | 110 | 156 | 131 |
| | SOS | FFT | | 75 | 149 | 139 |
| | MAX | | **199** | | | |
| | MAX | Sav-Gol | | 145 | 209 | 210 |
| | MAX | D-Log | | 162 | 235 | 192 |
| | MAX | Linear | | 159 | 188 | 169 |
| | MAX | FFT | | 163 | 191 | 217 |
| | EOS | | **284** | | | |
| | EOS | Sav-Gol | | 217 | 273 | 285 |
| | EOS | D-Log | | 258 | 269 | 285 |
| | EOS | Linear | | 250 | 266 | 286 |
| | EOS | FFT | | 249 | 271 | 274 |

To better explain the results, Table 6 shows the averages of the days of difference between the sensors and PhenoCams. The first three items correspond to the total average between the areas, while the last four items are divided based on the altitude.

**Table 6.** Average number of days of difference between sensors and PhenoCams.

| Sensor | Areas | SOS (day) | MAX (day) | EOS (day) |
|--------|-------|-----------|-----------|-----------|
| S-1 | all | 10 | 10 | 20 |
| S-2 | all | 4 | 10 | 8 |
| RSR | all | 1.5 | 18.5 | 2 |
| S-1 | 1500 | 9 | 1 | 18.5 |
| S-1 | 2000 | 14 | 19.5 | 21 |
| S-2 | 1500 | 1.5 | 8.5 | 4 |
| S-2 | 2000 | 9 | 12 | 11.5 |

Moreover, Figure 7 shows the DOY of the SOS extracted from each sensor (a) and percent error (b) on the SOS date estimation for all the areas. In Figure 7a, S-1 (light blue bars) detects the SOS before S-2 in stations at 1500 m, while it is delayed at 2000 m a.s.l. On the other hand, when comparing satellites with ground sensors, S-1 is less effective than S-2 in almost all stations, as illustrated in in Figure 7b. Only in the domef 2000 area, S-1 has a better result than S-2.

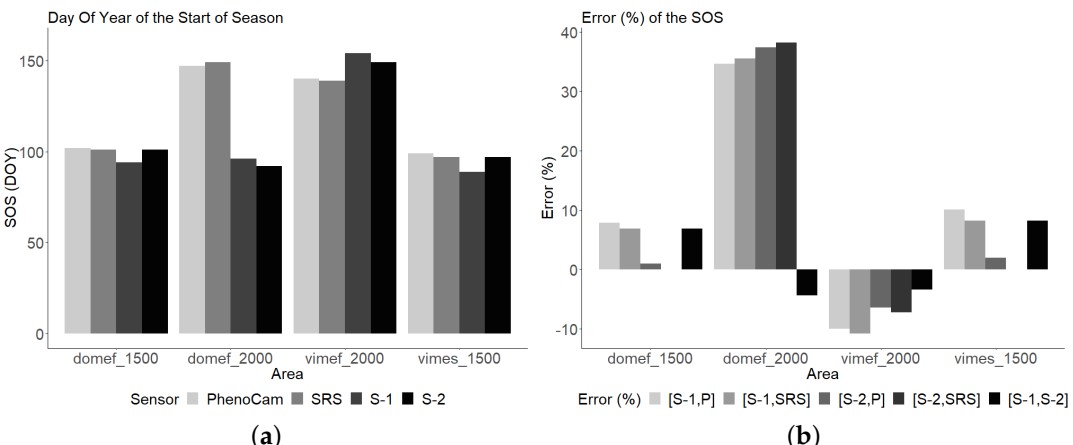

(**a**)　　　　　　　　　　　　　　　　　　　　　　　　(**b**)

**Figure 7.** DOY detected and percent errors between sensors in the analysis of the SOS: in (**a**) the different color bars represent the results for each sensor; in red are shown in (**b**) the percent difference between S-1-PhenoCams ([S-1,P]) and S-1-SRS ([S-1,SRS]), in yellow between S-2-PhenoCams ([S-2,P]) and S-2-SRS ([S-2,SRS]), while in orange the percent difference between S-1-S-2 ([S-1,S-2]).

Finally, we mapped the SOS with S-1 and S-2 time series, by using a linear filter. The first map is obtained from S-1 $\sigma_{VH}$ time series with a backscatter threshold of 0.9 (Figure 8a), and the second one from S-2 NDVI time series using a NDVI threshold of 0.7 (Figure 8b). Each class corresponds to a different SOS interval of time for a total of 10 classes, from DOY 61 to DOY 210. All non-vegetated areas are masked using CORINE 2012 land-cover information. Above a certain elevation though, as indicated in the statistical analysis, the $\sigma_{VH}$ loses the sensitivity to vegetation and gave unreasonable results. For this reason, the SAR map (Figure 8a,c) is masked above 2100 m of altitude. A detail of the map is shown in Figure 8c,d, where the station domef 1500 is located.

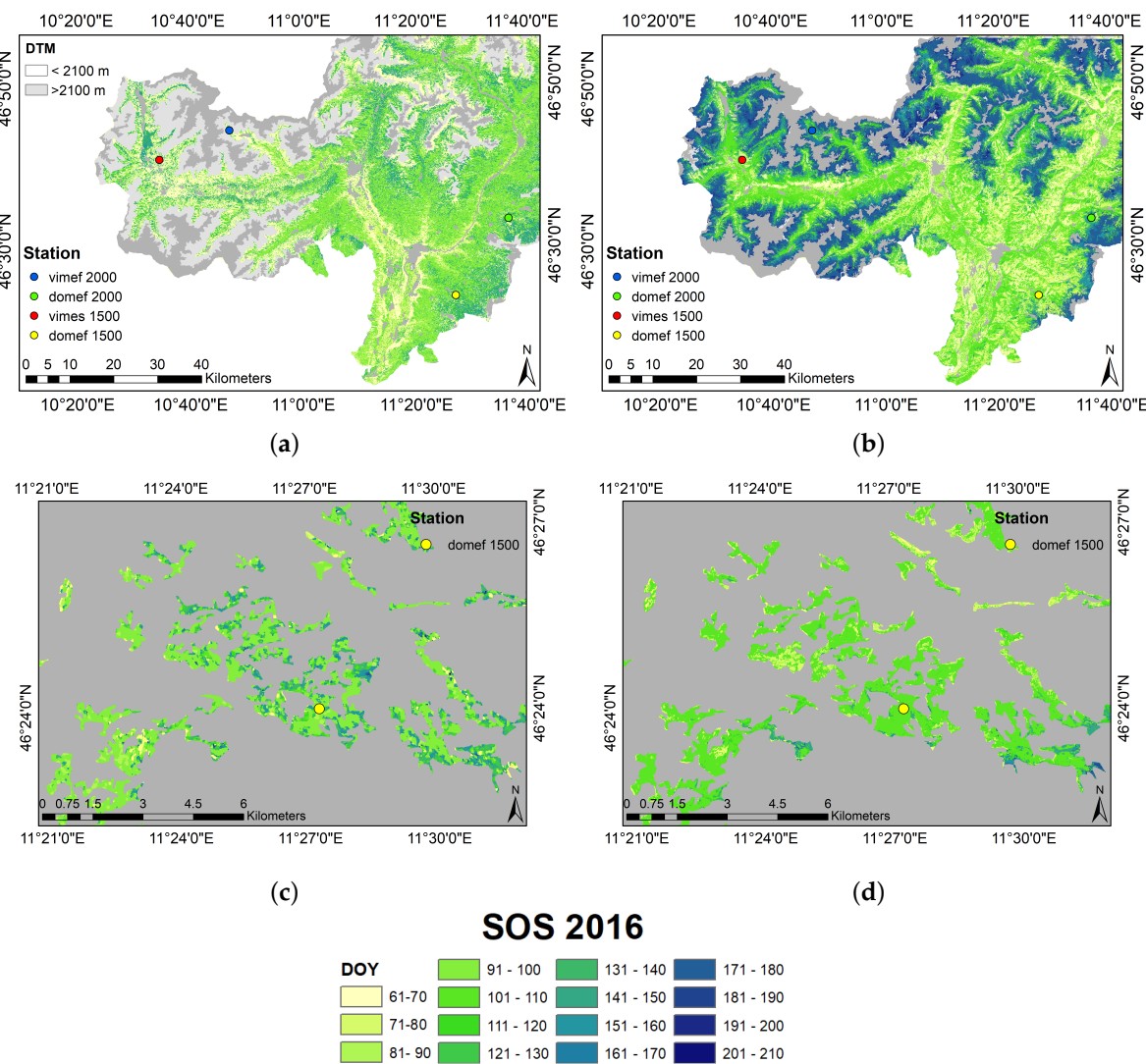

**Figure 8.** Time of start of the growing season in the South Tyrol region, using S-1 $\sigma_{VH}$ (**a**) and S-2 NDVI (**b**) time series. The SAR map is masked above 2100 m of altitude while only a non-vegetated area mask is applied on the optical one. In the bottom panel a detail of the maps of meadows areas around domef 1500 station, obtained with S-1 (**c**) and S-2 (**d**).

### 4.2.2. Harvest

In our study, first we automatically detected the harvest time in all the sensors, we compared them visually with images from PhenoCams, and finally we retrieve S-1 and S-2 harvest time maps. Figure 9 shows the detected timings of mowing for each sensor. In domef 1500 and vimes 1500 areas (Figure 9a,c) both optical sensors catch two mowing events; conversely, S-1 VH backscatter recognizes an individual event in between. In Figure 9d, S-2 detects two events instead of one due to a heavy snowfall corresponding to the day of acquisition of the optical satellite.

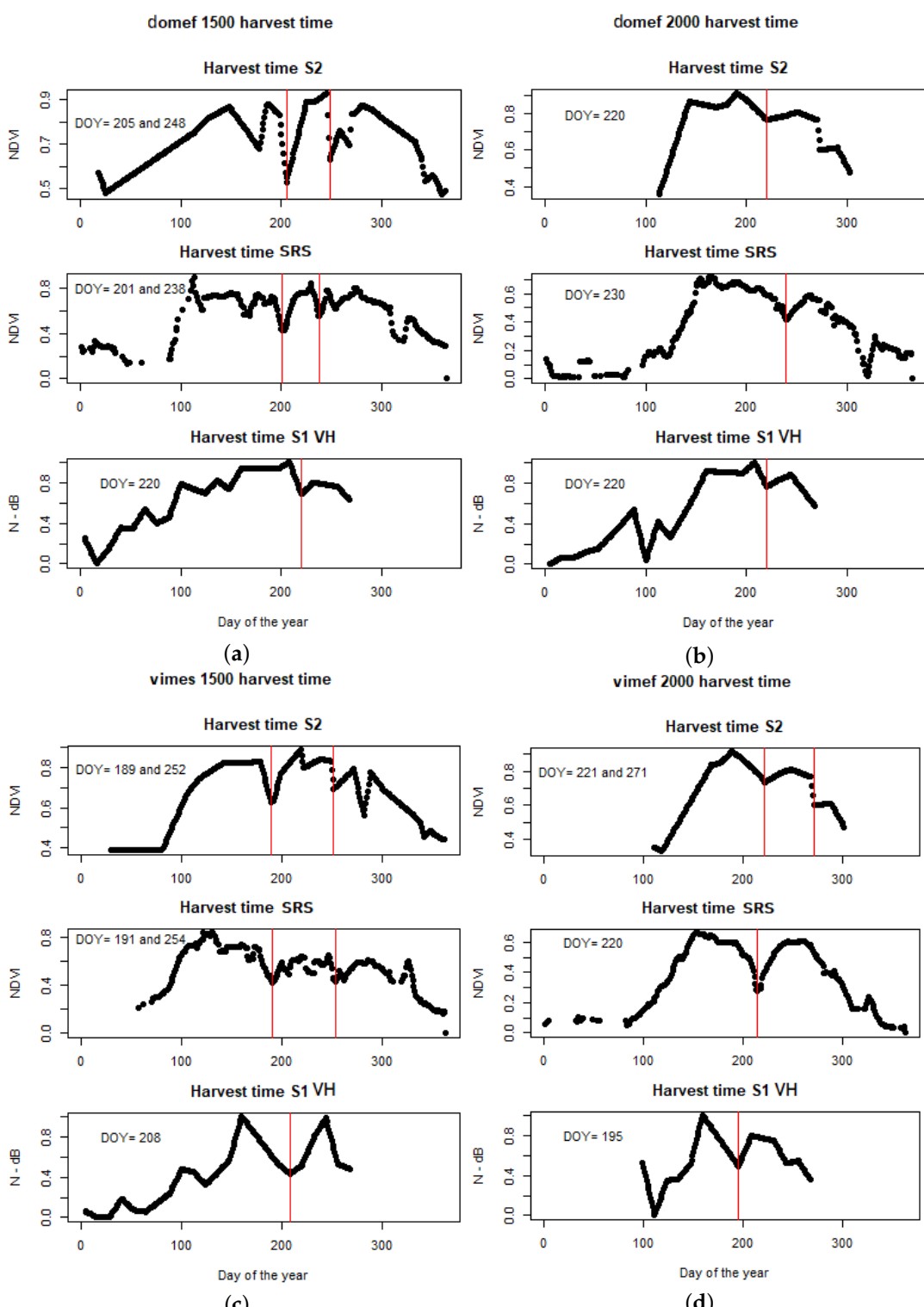

**Figure 9.** Detection of the harvest time (red line) using S-2, SRS, and S-1 time series. In the upper panel domef 1500 (**a**) and domef 2000 (**b**), in the bottom panel vimes 1500 (**c**) and vimef 2000 (**d**).

Table 7 illustrates the results from the S-1, S-2, and SRS compared with the harvest time detected from PhenoCams. S-1, as shown in Figure 10a, compared to both ground sensors and S-2, is delayed in the definition of the first mowing event, in low altitude stations. Conversely, S-1 is in advance in areas at 2000 m a.s.l., except for domef 2000, where S-1 and S-2 give the same result. In terms of percent error (Figure 10b), the results of S-1 are less accurate compared to S-2, except for the area domef 2000.



**Table 7.** Harvest time retrieved by PhenoCams, SAR and optical sensors.

| Area | Phenocam (DOY) | S-2 (DOY) | SRS (DOY) | S-1 (DOY) |
|------|----------------|-----------|-----------|-----------|
| domef 1500 | 198 and 252 | 205 and 248 | 201 and 238 | 220 |
| domef 2000 | 230 | 220 | 230 | 220 |
| vimes 1500 | 188 and 250 | 189 and 252 | 191 and 254 | 208 and 268 |
| vimef 2000 | 210 | 221 and 271 | 220 | 195 |

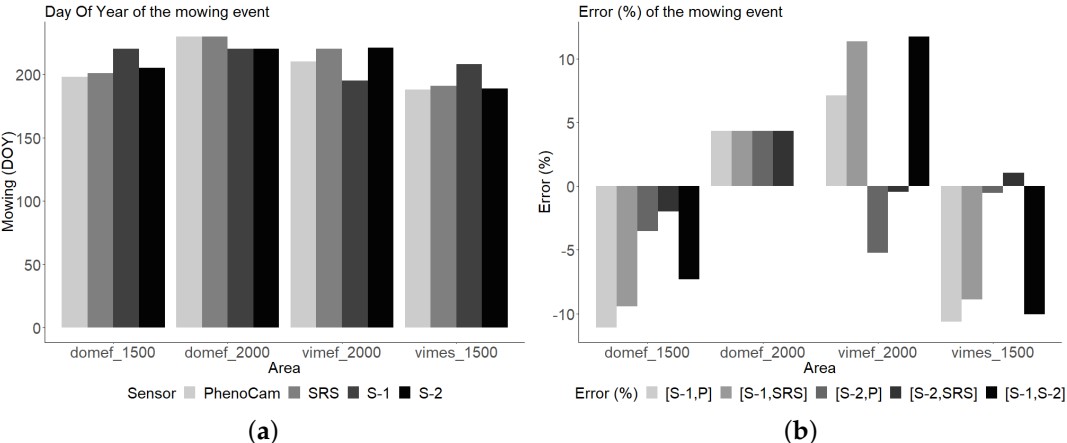

(**a**)                                    (**b**)

**Figure 10.** First mowing event detected by the different sensors: in (**a**) the different color bars represent the results of each sensor; in light blue are shown in (**b**) the percent error between S-1/PhenoCams ([S-1,P]) and S-1/SRS ([S-1,P]), in orange between S-2/PhenoCams ([S-2,P]) and S-2/SRS ([S-2,SRS]), while in purple the difference between S-1/S-2 ([S-1,S-2]).

Figure 11a,b illustrates the harvest time between DOY 180 and DOY 221 of meadow areas close to the station vimes 1500. The first map (a) is obtained from Sentinel-1 VH time series, while the second map (b) is generated from Sentinel-2 NDVI. The SAR map shows a harvesting time between DOY 201 and 220 for most of the meadow areas (green and yellow color); conversely, the optical map presents an earlier harvest among DOY 180 and 200 (pink and violet color). Except for the area where the mowing is beyond DOY 220, the two maps give a result that do not correspond, with a shift in time of the SAR result. SAR and optical time series give a corresponding result from DOY 221, as we can see in orange on the maps.

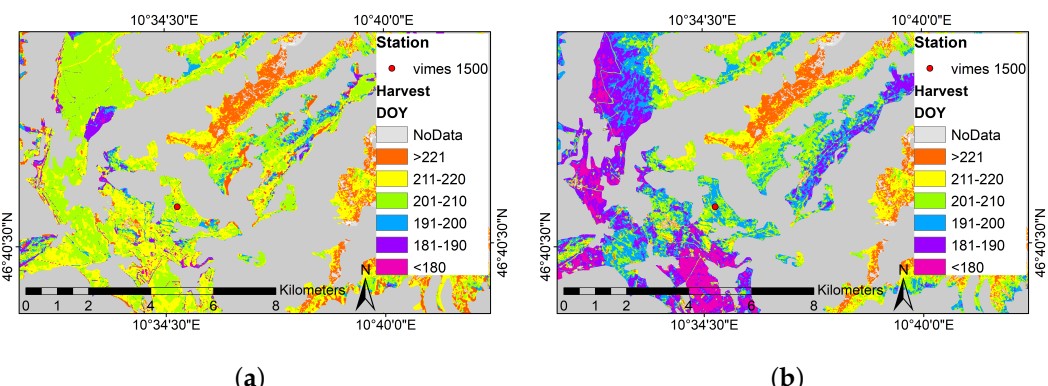

(**a**)                                    (**b**)

**Figure 11.** Harvest maps of the first mowing event generated from S-1 and S-2 (summer 2016). The maps show a detail of the surrounding vimes 1500 area: on the left panel (**a**) the map obtained from $\sigma_{VH}$ backscatter; on the right panel (**b**) the map obtained from Sentinel-2 NDVI.

## 5. Discussion

Our investigation revealed that time series of Sentinel-1 C-band allows phenological dynamics detection in different vegetation land-cover types. As shown in previous studies [58,59], VV and VH polarizations have a clear seasonal dynamic, with a peak that might correspond to the maximum of biomass production. It seems also possible to discriminate the signal of different vegetation classes. Moreover, the results achieved in the current study suggest that the backscatter of mountain meadows has a high dynamic range.

In pastures class, backscattering profiles are stable in VH polarization, while changing trend was observed in the VV channel. It might be possible that the contribution of bare soil and vegetation structure (short and thin leaves) for this class increases the sensitivity to variations in the water content of soil, instead of vegetation cover, as demonstrated in [60]. Furthermore, at high altitudes, there are limitations on SAR $\sigma^0$ VH sensitivity: the presence of low biomass with narrow leaves might increase the absorption effect, causing a flat trend in the backscatter coefficients [61].

Conversely, the VH channel better describes meadow phenology. As previously demonstrated [62], the correlation between $\sigma^0$ coefficients and NDVI is stronger in the VH channel in meadows areas, due to the volume scattering of vegetation. Similar results were found in [63], where $\sigma^0$ VH sharply raises during the phase of green-up, it is stable during the vegetation reproduction, and decreases rapidly due to the harvest. The cross-correlation between S-1 $\sigma^0$ VH backscatter and S-2 NDVI for the season 2016, shows a positive correlation for the selected areas. Moreover, the Pearson's product-moment correlation between S-1 and the SRS ground sensor reveals that in areas where the cloud cover limits S-2 data acquisition, backscattering coefficients can support the phenological phase detection. Our analyses were limited to one year, 2016. Since, time series contain a combination of seasonal, gradual and abrupt changes [64], a decomposition analysis should be applied to longer time series. A seasonal-trend analysis could therefore be useful in further studies when multiple years of Sentinel data will be available.

To derive useful quantitative information regarding the contribution of the vegetation to the SAR backscatter, we used the WCM. This semi-empirical model represents the power backscattered by the whole canopy as the incoherent sum of the contribution of the vegetation and soil [65]. Including NDVI in the model allows understanding the vegetation contribution to the VH channel. As emphasized by [66], in the VH channel, the vegetation contribution to the backscattering coefficient is higher than the soil component, when the vegetation is well developed [67].

The results of the comparison between the values predicted by the WCM and $\sigma^0$ VH time series show that the model is adequate to describe vegetation in mountainous areas. The statistical results are in line with previous studies for both Adjusted $R^2$ and Root-Mean-Square Error [68,69]. Moreover, the altitude does not seem to interfere with the simulation, giving a RMSE of 1.63 dB in an area at 2000 m a.s.l. Furthermore, through the analysis of $R^2$, $sigma_{VH}$ is more influenced by the vegetation growth than SWC. Hence, our results confirm that VH C-band SAR data combined with optical data may be applicable to estimate the vegetation phenology in mountain meadows.

To obtain the best mapping results, we evaluated different filter techniques, based on previous studies [70,71]. It is important to underline that in the validation phase, there are significant limitations in comparing satellite sensors and ground observation [72]. Whereas the NDVI is a direct measure of radiation absorption by the canopy [19], PhenoCam visual analysis, has different sources of uncertainties, especially to track when the first leaves appear from the surrounding vegetation and the mixture of senescence leaf colors [49,73,74]. For this reason, we evaluated the accuracy of our results through both the NDVI ground sensor (SRS) and PhenoCam images. The two results were in good agreement, with a mean of 1.5 day of difference for the SOS and 2 days for the senescence. Phenocams resulted essential to detect the harvesting time, by directly observing the mowing operations.

All four filters clearly describe the trend of the growing season in each area and none show better performance compared to the others. As expected, each filter applied to SRS NDVI time series

approximates well the seasonal phenology, even though, despite BISE noise-reduction techniques, the mowing events interfere with the detection of the EOS.

The days of difference of S-1 with respect to the dates extracted from the PhenoCams and SRS sensor increase with the altitude of the areas. For the SOS, at 1500 m a.s.l., the distance between the field data and the SAR data is compatible with the time of acquisition of the satellite. Conversely, in the areas at 2000 m, the distance in days exceeds the temporal resolution of the SAR satellite. The same tendency is repeated for the EOS, where, however, the difference increases with respect to the ground data. The optical data follows the trend of the SAR, with fewer days of difference. In this context, the percent error ranges between $-10\%$ in the worst scenario and 8% in the best one for S-1 and ground sensors; $-7\%$ and 2% of error respectively, for S-2 and ground sensors. In this analysis we do not consider the SOS extracted from both S-1 and S-2 time series in the area domef 2000; this exception is determined by the fact that in this area:

- a heavy snowfall in April, corresponding to the day of S-1 acquisition, caused a signal drop and consequently errors in filter modeling;
- in the optical domain, during the period January–October 2016, only 13 images were cloud free in this area.

Although the results of S-1 are in most cases less accurate than those of S-2, we expect that applying our detection method on flat areas and/or with different vegetation cover and leaves structure, we could have consistent results among SAR and optical sensors. Furthermore, we think that by increasing the temporal resolution, with the S-1B and S-2B acquisitions, the accuracy in the phenology estimation process would increase for both sensors.

In the mapping process, since from our comparative test the filters perform equally well, to have an identical approach in the optical and microwaves domains, and for simplicity reason, we applied a Linear Filter to both the time series. In both maps the growing season follows an altitude-based gradient, with an early start of vegetation growth at valley floors, anticipated in the wider areas, which is gradually delayed at high altitudes and especially in narrow valleys. For the vegetation that is covered by snow several months during the year, i.e., above 2500 m a.s.l., the green-up starts between the end of May (DOY 147) and the start of August (DOY 210). The map obtained from the $\sigma_{VH}$ shows less sensitivities at high altitude, where the vegetation decreases in height and biomass. Furthermore, the presence of bare soil strongly influences the SAR signal. The optical map shows an earlier start at the bottom of the valleys (around DOY 60–70), compared to the SAR detection (around DOY 80–90) and emphasizes the green-up gradient going from low to high altitude. When we zoomed in the map, the S-1 backscatter gave a delay in the SOS of around 10 days in some areas. However, the S-1 backscatter seems to be more sensitive than the S-2 NDVI, diversifying more SOS periods. The comparison between the SOS maps of South Tyrol, obtained from S-1 and S-2, illustrates that SAR data can be used to detect the onset of the growing season in meadow areas. However, as demonstrated in the vegetation type analyses, the same procedure is not applicable to pasture areas. In this class the $\sigma_{VH}$ time series, with a flat trend, does not allow the phenology detection. A sensitivity analysis of VV channel and the ratio VV/VH needs to be further investigated to understand a possible contribution to phenology detection of pasture class. In addition, both maps should be validated at different altitude and on different vegetation cover types (i.e., forest classes).

In mountain regions, there is a transition from fertilized to unfertilized meadows and pastures. Grasslands located at low altitudes or in the valley are usually mowed several times during the growing season. With increasing elevation, agriculture is less intensive, and the mountain meadows are mown once a year and mostly grazed in autumn [75]. Our areas of interest located at 1500 m a.s.l. are usually mowed twice a year, while those at 2000 m a.s.l. only once. Starting from the assumption that optical sensors well describe the radiation changes related to physiological conditions of plants, but they do not explain modification of the vegetation geometry [33], we expected to obtain a better harvest time detection with SAR time series. However, the S-1 GRD products were missing for the

season 2016 because of an onboard anomaly recorded between 8 June and 14 July [47]. This led to errors in the definition of the first mowing event. In terms of percent error, the results of S-1 are less accurate compared to S-2, with a range of error between −11% to 11% in the areas at lower altitude. Conversely, when there is only one mowing, in August at high altitudes, S-1 data give promising results (percent error between −8% to 4%) as well as S-2 (percent error between −5% to 4%). In these areas the mowing maps follow, indeed, the same trend. This demonstrates that S-1A instrument unavailability caused errors in the first mowing detection. Concurrently, the result suggests that in the presence of time series without missing data, S-1 gives results similar to S-2, allowing to overcome the problems of cloud cover in optical images. Having consistent data is indeed decisive in the definition of a mowing event. Furthermore, the advances/delays in the harvesting time detection are derived by the averaging of the selected areas which include different time of mowing. An example is shown in Figure 12 where in (a) at the top left we can see the start of mowing operations on DOY 220 and in (b) the end of them on DOY 235. Therefore, even in the case of mowing detection, using images from S-1B and S-2B, would improve our results.

Optical remote sensing provides a powerful tool to monitor phenology in mountain ecosystems and, our investigation has shown that SAR data might be effective in meadows phenology detection as well as complementary to the optical information. However, to test the applicability of the method on different vegetation classes more validation points are needed as well as a threshold's optimization. We cannot expect to obtain the same results in microwave and optical domains, due to different physical mechanisms: the first based on the structure, roughness, dielectric constant, and slope/orientation of scattering surfaces [22,76], and the second one on the reflectance properties of leaves, illumination angle, leaf orientation, and background [77]. In this context, our approach aims at understanding the behavior of the backscattering coefficients in meadow areas to complement the optical data with SAR images to reduce missing information caused by clouds contamination and atmospheric effects in the optical domain.

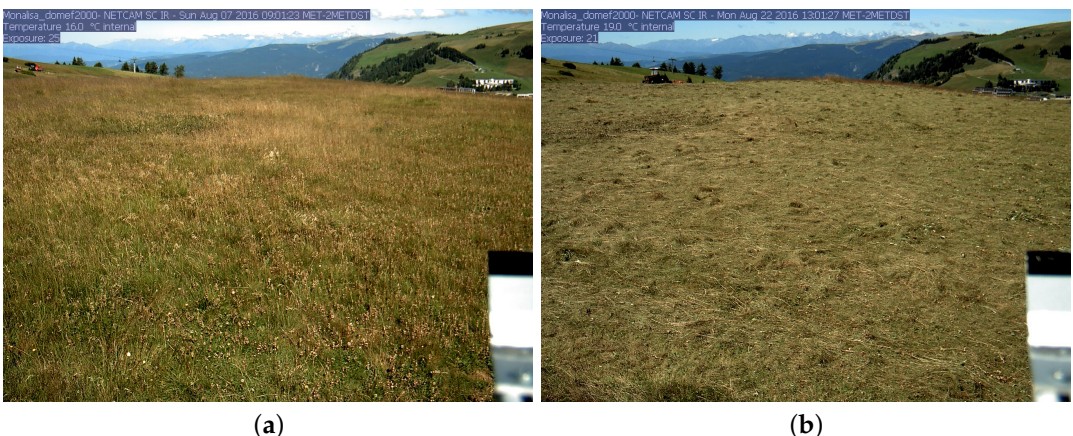

(**a**)                                          (**b**)

**Figure 12.** Images from the PhenoCam at the domef 2000 station. In (**a**) the DOY 220: at the top left the start of the mowing. In (**b**) DOY 235 the mowing right in front of the PhenoCam.

## 6. Conclusions

The paper describes multitemporal Sentinel-1 C-band and Sentinel-2 NDVI application on mountain meadows monitoring. The main aim was to test the feasibility of phenocycle phases retrieval from SAR time series and compare the results with the optical sensors, in the perspective of data integration.

From our analysis:

- The statistical analysis of $\sigma^0$ time series showed that the SAR signal can detect phenological cycles in different vegetation cover types.

- The significant correlation, with a negligible shift in lags, between $\sigma_{VH}$ and the NDVI from optical sensors, allowed the extraction of the phases of start, maximum and EOS, in addition to the mowing period.
- SAR data can be used to detect the phenological phases in meadows areas, with an accuracy compatible with the temporal resolution of S-1 until 1500 m a.s.l.

This result appears promising in the SAR-Optical data integration process for phenology detection. However, it needs to be confirmed for different altitudes and vegetation types. The data unavailability during the mowing period led to errors in the definition of the first harvest time. For this reason, future studies should be considered Sentinel-1B and Sentinel-2B acquisitions to increase the data consistency.

**Author Contributions:** All authors contributed extensively to the work presented in this paper. Project conceived and designed by C.N., L.S., G.N., S.R.K., R.G. and M.Z.; preprocessing of S-2 data by M.R.; Fieldwork designed and data acquired by G.N.; data analyses carried out by L.S. with support from S.R.K. and C.N.; Manuscript preparation lead by L.S., with support (substantial critical feedback, revisions and additions to text) from all co-authors; Final version of the manuscript read and approved by all co-authors.

**Funding:** This research received no external funding.

**Acknowledgments:** The station network is supported by the Autonomous Province of Bolzano/Bozen-South Tyrol within the frame of the projects MONALISA. We are grateful to Eurac research (Alessandro Zandonai and Stefano Della Chiesa) for field maintenance and Norut for the for logistical support and data preprocessing. We will also thank Carlo Marin for S-1 time series preprocessing. Finally, a grateful remark to Piyush Kumar for the language critical reading support.

**Conflicts of Interest:** The authors declare no conflict of interest.

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
