# Peer review of "Exploiting Time Series of Sentinel-1 and Sentinel-2 Imagery to Detect Meadow Phenology in Mountain Regions"

_remotesensing, doi:10.3390/rs11050542_

Round 1

Reviewer 1 Report

Abstract:

the authors must clarify what do they mean by a % error regarding the SOS, as it is now, it is confusing and not quite clear in the abstract. Also, the authors should edit the new section of the abstract to maintain the same structure as in the rest of the text (the new section seem odd compared to the rest of the manuscript).

Introduction:

L49: “A data fusion approach was developed 49 in a previous study “, the authors must place brief description of the approach.

Results:

For figures 7 and 10, the authors should consider the use of gray scales and pattern designs to present the bar graph, instead of bright colors.

Author Response

Dear Reviewer,

Thank you for your comments.

Please find attached the pdf with our response.

Best Regards

Reviewer 2 Report

It is generally ok to me, and I thank the authors for fully addressing my comments.

In Section 2.2, it should be pointed out that how many dates S-1 and S-2 data were collected, instead of just giving the satellite revising periods. For example, is there some exclusion because of cloud contamination? And I also wonder how the curves in figure 3 and figure 5 were generated considering the data points are very diverse.

Author Response

(The authors gave the same response as above.)

Reviewer 3 Report

I would suggest to accept the Ms since it has be well modified.

Author Response

Dear Reviewer,

Thank you for your comments.

Please find attached the pdf with our response.

Best Regards

This manuscript is a resubmission of an earlier submission. The following is a list of the peer review reports and author responses from that submission.

Round 1

Reviewer 1 Report

I am not in the field of vegetation phenology but I do very insterest in the work since I was doing landscape phenology (soil freeze-that cycles) detection with radar backscatter data. The paper is generally interest to scientists of ecology and geography. I would suggest to accept after a modertate revison. My concerns on the manuscript followed:

the validation in the manuscript. Especially the ground validation. Since the Ms forcus on grassland phenology, some times the ground measured phenology may later than the real phenology occur, for instance, in springtime, the grass litter is on the top of new grass, although the growth occured, no evidence could be observed by eyes. The author should indicate such kind of misleading and misjudgement in the data and valication. A comphensive disscussion on this point should be given in the text. 

The reasons that made VV and HH change when certain phenology occured should be given, for instance, if soil thaw occured, the soil delective constant will be changed and thus made the rada backscatter change. Such theory for vegetation phenologoy change is highly suggested to give in the method of introduction section. 

the title of the Ms refers to "grassland", while in the Ms, author mention the key points is AGRICULTURAL. Such things would made readers confused on what topic the author carred this research. This point should be clear before acceptance. 

Maps in the Ms should be more readable and beautiful.

conclusion should give distinct bullets to highlight the major results from this research. 

Reviewer 2 Report

As shown in Figure 5, the measured VH has bad linear correlation with WCM and SWC in the four station areas.

On page 10, Line 261, the word “schow” should be “show”. In addition, on page 13, in Table 5, in the fifth row, the number 826 is a typing error?

As pointed out in the manuscript, the results extracted from S-1 and S-2 are different due to the two types of remote sensing mechanism, however, compared with the ground truths, which is better? Some statistical analysis should be conducted to validate the results, which is a bit weak in the current manuscript.

It should be addressed the advantages of using S-1 compared to the optical one S-2 in the SOS and harvest date extraction for different vegetation classes and different topographic locations.

Figure 1 is not necessary, but a DEM and land cover map might be added to help understand the results.

The innovation of the method shold be explicitly addressed in the end of Section introduction or the conclusion section.

Reviewer 3 Report

The paper provides an analysis of the ability of Sentinel-1 and Sentinel-2 Synthetic Aperture Radar (SAR) to track changes in alpine non forested ecosystems phenology. The motivations behind this project are solid. Monitoring phenology with traditional optical sensors is challenging in case of intense cloud cover, while methods using SAR may not. 

The authors (1) explore the trends of backscattering in different vegetation cover types, (2) evaluate the cross correlation between SAR backscattering and NDVI, Temperatures, and Soil Water Content dynamics, (3) they simulate SAR signal with a Water Cloud Model to decouple the effects of soil and vegetation on the observed signal. They extract the dates of start, maximum and end of vegetative phase by filter the SAR curves, and compare them with PhenoCam data. 

This work is useful, the authors gathered very good references, but I am partly concerned about it being mainly an application of known techniques on a newer dataset. Moreover I am partially concerned on if the highly seasonal nature of the data could lead to statistical "artifacts" if not normalized.

I loved the presence of the links to webpages with more information on the ground stations and the MONALISA project!

Major concerns

The paper is good in length, but may benefit from a bit of restructuring and rewriting. Some sentences are too long and/or complex. 

Captions in the tables need more information. Would be beneficial to add units when possible (e.g. RMSE in table 3), and redefine acronyms. 

I got confused by how the comparison between the three time series are used to provide a sensitivity analysis. From the term "sensitivity analysis" I was expecting the analysis of the variation of some models' outputs by changing parameters values and ranges. If I am not mistaken, the authors are mainly exploring the covariation between VH, NDVI, Temperature and Moisture trends. In the analysis the authors provide the Pearson correlation coefficient. They briefly mention the presence of negligible lags; I think structure of lags would be a useful information to provide (maybe in the supplements). 

Another concern is that the correlation coefficients may be affected by seasonal trends, and have high values for statistical rather than physical reason. It would be useful in this perspective to show all the analyses done, and possibily normalize the time series to get rid of the seasonality.

I found very interesting the use of the WCM to decouple the effects of NDVI and SWC on the backscattering coefficient. I think the authors should better describe how they analized the outputs of the optimized model in the methods section.

Minor concerns

I suggest to reconsider the choice of some terms. For example, in line 109, the authors refer to the 4 studied areas as to "test areas". The term confused me at first, because it generally implies a split of the dataset in train and test, with the former used to build a model, the latter to validate it on independent data. 

Could you add a reference or a brief description in the methods section for the cross-correlation function used?

The results section is probably too long and some portions would benefit from being addressed in the discussion (for example lines 277 to 295, or most of section 4.2.2). This may be just a style minor issue, but I think would make the article flow easier.

The project is definitively interesting, and well suited for Remote sensing. I think though that as it is, it is hard for the reader to follow the flow and grasp its take away message. 

Hope these comments will have been of any use to make your manuscript better.

Best regards, and good luck!

Reviewer 4 Report

Title: Exploiting time series of Sentinel-1 and Sentinel-2 imagery to detect grassland phenology in mountain regions

Abstract:

The abstract presents a proper organization of the sections. However, I would recommend to be more specific when describing results.

This needs to be clarified:  “From our analyses, Sentinel-1 detect the Start Of Season on meadow areas with an average of 6 days difference compared to Sentinel-2 NDVI retrieval and 12 days with respect to the End Of Season”. What are the authors referring to with “6 days difference”? Before the SOS? After? What are the implications?

Also, it would be good to mention how “well correlated” are the VI´s and radar measurements (statistical range).

Introduction:

L43: the Idea must be completed: “results in…?”

L49: “phenology and parameters” did the author mean “phenology and its parameters?”

L72-76: I do not think this paragraph is necessary, I would recommend to delete it.

The intent is well delineated in the intro.

Study area and datasets:

Datasets:

The temporal resolution of the S-1 and S-2 images should be mentioned.

Methodology:

136: “(CLC 2012)” this is not clear, the authors should place a reference and not just the acronym.

Figure 3 should be relocated to its corresponding section.

Results: the section corresponds to the methodological proposal. However, it is necessary to highlight the relationships between the Optical and the Radar products more formally (not just an R2 but the significance and a mention to the test).